# A 2-Gene Host Signature for Improved Accuracy of COVID-19 Diagnosis Agnostic to Viral Variants

Jack Albright,[a] Eran Mick,[a,b,c] Estella Sanchez-Guerrero,[b] Jack Kamm,[a*] Anthea Mitchell,[a,d] Angela M. Detweiler,[a] Norma Neff,[a] Alexandra Tsitsiklis,[b] Paula Hayakawa Serpa,[b] Kalani Ratnasiri,[a] Diane Havlir,[e] Amy Kistler,[a] Joseph L. DeRisi,[a,d] Angela Oliveira Pisco,[a] ⓘ Charles R. Langelier[a,b]

aChan Zuckerberg Biohub, San Francisco, California, USA
bDivision of Infectious Diseases, Department of Medicine, University of California San Francisco, San Francisco, California, USA
cDivision of Pulmonary and Critical Care Medicine, Department of Medicine, University of California San Francisco, San Francisco, California, USA
dDepartment of Biochemistry and Biophysics, University of California San Francisco, San Francisco, California, USA
eDivision of HIV, Infectious Diseases and Global Medicine, Department of Medicine, University of California San Francisco, San Francisco, California, USA

Jack Albright, Eran Mick, and Estella Sanchez-Guerrero contributed equally to this article. Author order was determined alphabetically.

**ABSTRACT** The continued emergence of SARS-CoV-2 variants is one of several factors that may cause false-negative viral PCR test results. Such tests are also susceptible to false-positive results due to trace contamination from high viral titer samples. Host immune response markers provide an orthogonal indication of infection that can mitigate these concerns when combined with direct viral detection. Here, we leverage nasopharyngeal swab RNA-seq data from patients with COVID-19, other viral acute respiratory illnesses, and nonviral conditions ($n$ = 318) to develop support vector machine classifiers that rely on a parsimonious 2-gene host signature to diagnose COVID-19. We find that optimal classifiers include an interferon-stimulated gene that is strongly induced in COVID-19 compared with nonviral conditions, such as *IFI6*, and a second immune-response gene that is more strongly induced in other viral infections, such as *GBP5*. The *IFI6*+*GBP5* classifier achieves an area under the receiver operating characteristic curve (AUC) greater than 0.9 when evaluated on an independent RNA-seq cohort ($n$ = 553). We further provide proof-of-concept demonstration that the classifier can be implemented in a clinically relevant RT-qPCR assay. Finally, we show that its performance is robust across common SARS-CoV-2 variants and is unaffected by cross-contamination, demonstrating its utility for improved accuracy of COVID-19 diagnostics.

**IMPORTANCE** In this work, we study upper respiratory tract gene expression to develop and validate a 2-gene host-based COVID-19 diagnostic classifier and then demonstrate its implementation in a clinically practical qPCR assay. We find that the host classifier has utility for mitigating false-negative results, for example due to SARS-CoV-2 variants harboring mutations at primer target sites, and for mitigating false-positive viral PCR results due to laboratory cross-contamination. Both types of error carry serious consequences of either unrecognized viral transmission or unnecessary isolation and contact tracing. This work is directly relevant to the ongoing COVID-19 pandemic given the continued emergence of viral variants and the continued challenges of false-positive PCR assays. It also suggests the feasibility of pan-respiratory virus host-based diagnostics that would have value in congregate settings, such as hospitals and nursing homes, where unrecognized respiratory viral transmission is of particular concern.

**KEYWORDS** COVID-19, diagnostics, classifier, gene expression, metagenomics, transcriptomics

The COVID-19 pandemic has inflicted unprecedented human health consequences, with millions of deaths reported worldwide since December 2019 (1). Testing is a cornerstone of pandemic management, yet existing assays suffer from accuracy limitations. Even the

Address correspondence to Charles R. Langelier, chaz.langelier@ucsf.edu.

*Present address: Jack Kamm, Genentech, Inc., South San Francisco, California, USA.

The authors declare no conflict of interest.

gold-standard testing modality of nasopharyngeal (NP) swab RT-PCR returns falsely negative in a substantial proportion of cases (2 to 4) and may fail to detect SARS-CoV-2 variants with mutations at primer target sites (5 to 7). False-positive tests due to sample cross-contamination in the laboratory are also a significant complication (8, 9) as they can lead to costly contact tracing efforts and the unnecessary isolation of uninfected individuals, including essential workers.

Measuring the host immune response offers a complementary approach to direct detection of the SARS-CoV-2 virus and holds potential for overcoming the limitations of existing COVID-19 diagnostics. RNA-sequencing (RNA-seq) studies of NP swabs and blood have demonstrated that COVID-19 elicits a unique host transcriptional response compared with nonviral and other viral acute respiratory illnesses (ARIs) (10 to 12). A host gene expression signature of COVID-19, when utilized in combination with molecular detection of SARS-CoV-2, can serve as a fallback to identify suspected false-negative or false-positive results of traditional viral PCR tests, thus improving overall diagnostic reliability.

Recent studies have employed machine learning on RNA-seq data from NP swabs to develop host-based COVID-19 diagnostic classifiers that rely on a relatively large number of genes (10, 13). While highly promising, these classifiers have yet to undergo validation in external cohorts. Furthermore, RNA-seq is not widely available in clinical settings, and thus the immediate practical utility of RNA-seq classifiers is limited.

Here, we address these gaps by identifying 2-gene host signatures that could practically be incorporated into an RT-qPCR (qPCR) assay alongside a control gene and one or more viral targets. We leverage NP swab RNA-seq data from two large patient cohorts to derive and validate top-performing support vector machine (SVM) binary classifiers that use 2 host genes to diagnose COVID-19. The optimal 2-gene signatures combine an interferon-stimulated gene (ISG) that is strongly induced in COVID-19, such as *IFI6*, with another immune response gene that is more strongly induced in other viral ARIs, such as *GBP5*. We then provide proof-of-concept demonstration that such a 2-gene classifier can practically be applied to qPCR data using a third sample cohort. Finally, we show that the host classifier is robust across SARS-CoV-2 variants, including those that can yield a false-negative viral PCR result, and is unaffected by laboratory cross-contamination that can yield a false-positive viral PCR result.

## RESULTS

**Development and validation of a 2-gene, host-based COVID-19 classifier from NP swab RNA-seq data.** We previously developed multigene host classifiers for COVID-19 using RNA-seq data from NP swabs of patients tested for COVID-19 at the University of California, San Francisco (UCSF) who were diagnosed with either COVID-19, other viral ARIs, or nonviral ARIs (10). In the present work, we sought to develop a parsimonious 2-gene signature that could be practically incorporated into a PCR assay alongside a control gene and one or more viral targets.

We began by identifying top-performing 2-gene candidates in our RNA-seq cohort after supplementing it with additional samples collected in the intervening time. The full UCSF cohort used in the present work included $n = 318$ patients, of whom 90 had PCR-confirmed COVID-19 (with viral load equivalent to PCR cycle threshold ($C_T$) < 30), 59 had other viral infections detected by metagenomic sequencing (mostly rhinovirus and influenza), and 169 had no virus detected and were presumed to suffer from nonviral ARIs (Table S1; Data Set S1).

The UCSF samples were split into a training set (70%) and a testing set (30%), with stratification to ensure each one contained similar proportions of patients with COVID-19. We then applied a greedy selection algorithm to identify 2-gene combinations that best distinguished the patients who had COVID-19 from the patients who did not, regardless of whether they had another viral infection or no viral infection. The performance metric was the area under the receiver operating characteristic curve (AUC) of a support vector machine (SVM) binary classifier that used the selected genes as features, calculated using 5-fold cross-validation within the training set (Fig. 1a). Thus, a first gene was selected to maximize the AUC it achieved on its own, and a second gene was selected to maximize the AUC when combined with the first gene. Table 1 lists nine combinations consisting of each of the three

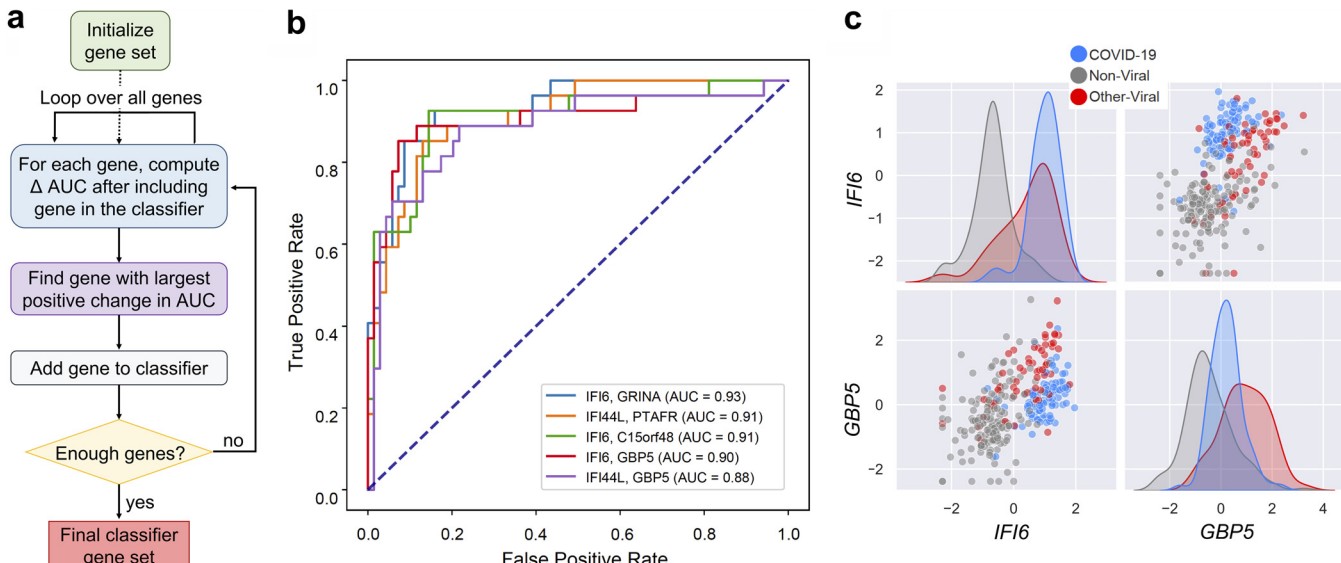

**FIG 1** Development of 2-gene host-based SVM COVID-19 diagnostic classifiers from RNA-seq data. (a) Schematic of the greedy feature selection algorithm used to identify top performing 2-gene combinations. (b) Receiver operating characteristic (ROC) curve demonstrating performance of SVM classifiers using the indicated 2-gene combinations. The classifiers were trained on the UCSF training set and applied to the UCSF testing set. AUC = area under the ROC curve. (c) Expression distributions of the "first" and "second" genes *IFI6* and *GBP5*, respectively, in the full UCSF cohort. Shown are variance-stabilized gene expression values after centering and scaling. Color indicates patient group.

best "first" genes and their respective three best "second" genes. The "first" genes in the top combinations were the interferon-stimulated genes (ISGs) *IFI6*, *IFI44L*, and *HERC6*, which we previously showed are strongly induced in COVID-19 (10). Most of the "second" genes were also related to immune and inflammatory processes.

The performance of the nine 2-gene combinations on previously unseen data was estimated by (i) 10,000 rounds of 5-fold cross-validation within the training set and by (ii) training on the training set and classifying the testing set (Table 1; Data Set S2). Using the latter approach, we observed AUC values as high as 0.93 (Fig. 1b; Table 1). On Youden's index, the top performing combinations achieved sensitivity in the range of 82 to 89%, positive predictive value (PPV) as high as 82%, and negative predictive value (NPV) as high as 95%. Overall specificity was as high as 93% (Table 1), though specificity with respect to the other viral samples posed a more significant challenge than specificity with respect to the nonviral samples. The best-performing combination in this regard was *IFI6+GBP5*, which achieved specificity of 96% with respect to nonviral samples and 80% with respect to other viral samples.

We further validated the classifiers using an external, independently generated and quantified NP swab RNA-seq data set from a cohort of *n* = 553 patients in New York (166 with COVID-19, 79 with other viral infections, 308 with nonviral conditions) (12) (Table S1;

**TABLE 1** Performance of the indicated 2-gene SVM classifiers for COVID-19 diagnosis in the UCSF RNA-seq cohort[a]

| 2-gene combination | 70% training set (*n* = 222) AUC 10,000 rounds of 5-fold CV | 30% testing set (*n* = 96) AUC Trained on 70% training set | PPV | NPV | Sens | Spec All | Spec No Virus | Spec Other Virus |
|---|---|---|---|---|---|---|---|---|
| IFI6, GRINA | 0.959 (0.005) | 0.934 | 0.686 | 0.951 | 0.889 | 0.841 | 0.907 | 0.600 |
| IFI6, C15orf48 | 0.949 (0.005) | 0.908 | 0.706 | 0.952 | 0.889 | 0.855 | 0.944 | 0.533 |
| IFI6, GBP5 | 0.948 (0.005) | 0.905 | 0.815 | 0.928 | 0.815 | 0.928 | 0.963 | 0.800 |
| IFI44L, GBP5 | 0.944 (0.004) | 0.883 | 0.605 | 0.931 | 0.852 | 0.783 | 0.852 | 0.533 |
| IFI44L, PTAFR | 0.934 (0.006) | 0.910 | 0.710 | 0.923 | 0.815 | 0.870 | 0.963 | 0.533 |
| IFI44L, FCGR1A | 0.932 (0.004) | 0.859 | 0.731 | 0.886 | 0.704 | 0.899 | 0.981 | 0.600 |
| HERC6, TNIP3 | 0.923 (0.005) | 0.844 | 0.714 | 0.897 | 0.741 | 0.884 | 0.926 | 0.733 |
| HERC6, GBP5 | 0.917 (0.005) | 0.841 | 0.633 | 0.879 | 0.704 | 0.841 | 0.907 | 0.600 |
| HERC6, C0A3 | 0.914 (0.005) | 0.816 | 0.571 | 0.885 | 0.741 | 0.783 | 0.889 | 0.400 |

[a]The area under the curve (AUC) is reported as mean and standard deviation when multiple cross-validation (CV) rounds were performed within the training set, or as a single score when the model was trained on a training set and evaluated on a testing set. The positive predictive value (PPV), negative predictive value (NPV), sensitivity (Sens), and specificity (Spec) values shown are calculated on Youden's index when the model is evaluated on the testing set.

**TABLE 2** Performance of the indicated 2-gene SVM classifiers for COVID-19 diagnosis in the New York RNA-seq cohort[a]

| 2-gene combination | External dataset (n = 553) AUC Trained on UCSF 70% training set | PPV | NPV | Sens | Spec All | Spec No Virus | Spec Other Virus |
|---|---|---|---|---|---|---|---|
| IFI6, GRINA | 0.883 | 0.725 | 0.908 | 0.795 | 0.871 | 0.922 | 0.671 |
| IFI6, C15orf48 | 0.861 | 0.699 | 0.897 | 0.771 | 0.858 | 0.912 | 0.646 |
| IFI6, GBP5 | 0.910 | 0.742 | 0.924 | 0.831 | 0.876 | 0.916 | 0.722 |
| IFI44L, GBP5 | 0.919 | 0.749 | 0.929 | 0.843 | 0.879 | 0.919 | 0.722 |
| IFI44L, PTAFR | 0.894 | 0.659 | 0.942 | 0.886 | 0.804 | 0.877 | 0.519 |
| IFI44L, FCGR1A | 0.896 | 0.701 | 0.929 | 0.849 | 0.845 | 0.883 | 0.696 |
| HERC6, TNIP3 | 0.852 | 0.687 | 0.915 | 0.819 | 0.840 | 0.899 | 0.608 |
| HERC6, GBP5 | 0.866 | 0.677 | 0.900 | 0.783 | 0.840 | 0.883 | 0.671 |
| HERC6, C0A3 | 0.797 | 0.563 | 0.898 | 0.807 | 0.731 | 0.795 | 0.481 |

[a]The positive predictive value (PPV), negative predictive value (NPV), sensitivity (Sens), and specificity (Spec) values shown are calculated on Youden's index when the model is evaluated on the testing set.

Data Set S1). The 2-gene combinations achieved comparable performance on the external data set (Table 2; Data Set S2). The best-performing combinations were *IFI44L+GBP5* (AUC 0.919) and *IFI6+GBP5* (AUC 0.91), when the classifier was trained on the UCSF 70% training set. On Youden's index, the classifiers achieved sensitivity of 83 to 84%, PPV as high as 74%, NPV as high as 92%, and overall specificity of 88% (Table 2). We observed that COVID-19 samples with very low viral loads were more likely to be misclassified as negative for COVID-19 (Fig. S1). These results demonstrate that 2-gene diagnostic classifiers for COVID-19 are feasible, generalizable, and perform well on real-world cohorts that include patients with other respiratory viral infections.

Among the top 2-gene combinations nominated by the greedy selection algorithm, *IFI6+GBP5* appeared to provide the best balance of sensitivity and specificity with respect to both nonviral and other viral samples. When visualizing the expression of this gene pair in the UCSF cohort, we noted that *IFI6* alone almost completely separated the COVID-19 and nonviral samples (Fig. 1c). However, some of the other viral ARI samples showed equivalent levels of *IFI6* expression. Adding *GBP5* allowed for improved separation, as expression of this ISG was typically higher in other viral ARIs (Fig. 1c). We confirmed that in both the UCSF and the New York cohorts, these genes exhibited fold changes between patient groups that should be detectable by qPCR (*IFI6* COVID-19 versus no virus $\log_2$FC ~4; *GBP5* COVID-19 versus other virus $\log_2$FC ~1.5), and so this pair was chosen for implementation in a proof-of-concept qPCR assay.

**Proof-of-concept implementation of a 2-gene, host-based qPCR COVID-19 diagnostic classifier.** Having validated the performance of the host-based classifier using the RNA-seq cohorts, we sought to demonstrate it could technically be implemented in a clinically relevant qPCR assay. We therefore measured the expression of *IFI6* and *GBP5* (relative to the reference gene *RPP30*) using qPCR assays on swabs from a new cohort of patients with (n = 72) or without (n = 72) COVID-19 (Table S2; Data Set S3). Because these swabs were not sequenced, we could not definitively assign those without COVID-19 as either nonviral or other viral cases. However, the low prevalence of other viral ARIs during the time frame of sample collection, due to the public health measures implemented for COVID-19 (14), suggested they were mostly nonviral. Using 5-fold cross-validation, we observed that the *IFI6+GBP5* SVM classifier achieved an AUC of 0.842 (±0.08) in distinguishing patients with and without COVID-19 from the qPCR data (Fig. 2a; Table S3).

**Host signatures are robust to SARS-CoV-2 variants and laboratory cross-contamination.** We next assessed whether the 2-gene host classifier was robust across SARS-CoV-2 variants, which could conceivably yield an altered host response and/or harbor mutations that disrupt primer target sites and lead to false-negative viral PCR tests (5, 7, 15). We performed qPCR for the genes *IFI6* and *GBP5* on samples with the Omicron variant (n = 3), which causes S-gene target dropout in certain viral PCR assays; on samples with the California N-gene variant (n = 4), which causes N-gene target dropout (15); and on samples with the Delta variant (n = 7). SVM classifiers trained on the qPCR results of the samples with and without COVID-19, described above,

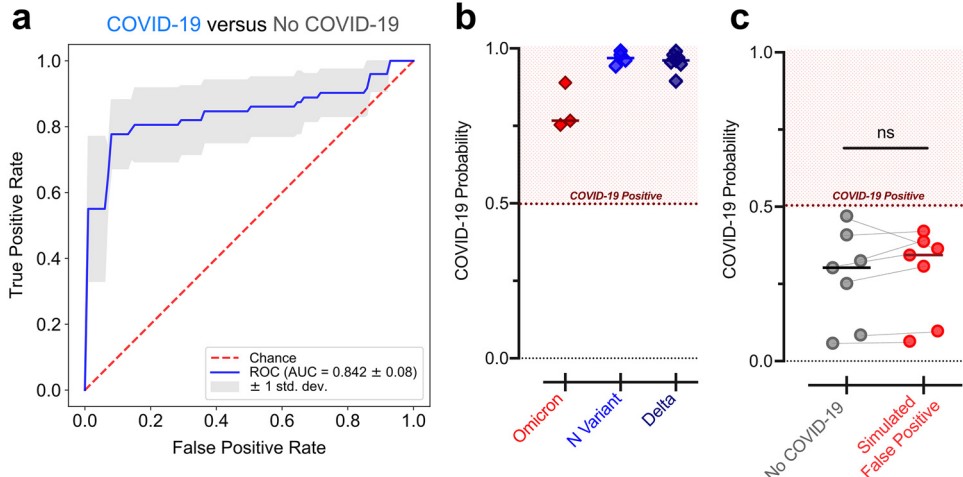

**FIG 2** Performance of 2-gene SVM COVID-19 diagnostic classifiers in qPCR assays. (a) ROC curve demonstrating performance of the *IFI6*+*GBP5* SVM classifier for distinguishing samples from patients with and without COVID-19 using the qPCR data, estimated by 5-fold cross-validation. Mean and standard deviation of the AUC across the five folds is reported. (b) Average probability of COVID-19 derived from the *IFI6*+*GBP5* qPCR cross-validation classifiers for samples with the Omicron variant (*n* = 3), the N-gene variant (*n* = 4), and the Delta variant (*n* = 7). (c) Average probability of COVID-19 derived from the *IFI6*+*GBP5* qPCR cross-validation classifiers for *n* = 7 samples without COVID-19 before and after introduction of trace contamination from a sample with high SARS-CoV-2 viral load. Statistical significance was assessed using a one-sided (greater than) paired Mann-Whitney test. ns, not significant.

predicted COVID-19 with high likelihood in all variant samples (Fig. 2b), demonstrating the potential utility of a host signature as a complement to viral PCR.

On the other hand, false-positive viral PCR tests frequently result from trace cross-contamination of samples with high viral titers into negative specimens processed contemporaneously in the laboratory (9). To examine whether an *IFI6*+*GBP5* host classifier would also be affected in such cross-contamination events, we spiked extracted NP swab RNA from a sample with very high SARS-CoV-2 viral load ($C_T \approx 12$) into *n* = 7 COVID-19 negative swab specimens at a dilution of $1:10^5$, which would be expected to yield a positive viral PCR result with $C_T < 30$. Reassuringly, however, the host-based probability of COVID-19 was not significantly affected in the contaminated specimens (Fig. 2c).

## DISCUSSION

We leveraged multiple cohorts—encompassing over 1,000 patients with COVID-19, other viral ARIs, and nonviral conditions—to develop and validate 2-gene host-based COVID-19 diagnostic classifiers that could be practically incorporated into clinical PCR assays in combination with a control gene and one or more viral targets. We found that the host classifier enabled reliable identification of COVID-19 even in the face of SARS-CoV-2 variants that cause false-negative viral PCR tests, and remained unaffected by simulated laboratory cross-contamination that can cause false-positive viral PCR tests.

Given the inevitable continued emergence of SARS-CoV-2 variants, which may disrupt primer target sites, assays capable of detecting infection regardless of viral sequence are essential to avoid adverse outcomes owing to infected individuals going unrecognized in congregate settings, such as hospitals or nursing homes. The adverse effects of false-positive tests are also nontrivial. The positive predictive value of highly specific viral PCR assays diminishes for asymptomatic individuals undergoing continual surveillance testing in low prevalence settings (9). False-positive results then become more likely, leading to unnecessary isolation and quarantine, depletion of essential personnel, and unwarranted contact tracing.

Our host-based classifiers were developed and evaluated using the practical gold-standard of clinical viral PCR, which would be more accurate in the general case than any host-based classifier (at least for existing variants). We emphasize, however, that we do not envision the use of host classifiers as a replacement for viral PCR, but rather as a complementary approach to compensate for its potential failure modes. While our proof-of-concept work

suggests that addition of host targets is likely to improve overall diagnostic accuracy, a prospective assessment using clinically confirmed false-positive and false-negative viral tests is needed, and a randomized controlled trial of our assay will be required to firmly establish its clinical utility.

Our study has some limitations. Our classifier models were trained and tested on cohorts with particular characteristics, including the distribution of COVID-19, other viral, and nonviral cases; the mix of other respiratory viruses represented; and within the COVID-19 group, the distributions of viral load, time since onset of infection, and disease severity. Most of these variables likely affect classifier performance and will vary in reality with time and place. Moreover, the analyzed nasopharyngeal swabs represented a convenience sample derived from a clinical SARS-CoV-2 testing laboratory, as well as from publicly available data, which could introduce bias. However, the fact that our classifiers translated well across diverse real-world cohorts argues that they are quite robust to these issues. Finally, it is possible that host gene expression would differ in response to infection with future SARS-CoV-2 variants, which could impact host-based diagnosis.

While we did not explicitly explore it here, our results suggest that parsimonious host classifiers could serve not only as a COVID-19 diagnostic but also as a pan-respiratory virus surveillance tool. Even prior to the COVID-19 pandemic, viral lower respiratory tract infections were a leading cause of disease and death (16), and many respiratory viral infections go undetected, leading to preventable transmission and unnecessary antibiotic treatment (17). Since our classifiers rely heavily on ISGs and type I interferon signaling is a biologically conserved mechanism, these genes could be used in future work as the basis for a diagnostic that identifies respiratory viruses more generally. Such a diagnostic could have considerable value as a screening tool in hospitals, nursing homes, or other congregate settings with potential for adverse consequences from unrecognized respiratory viral transmission.

## MATERIALS AND METHODS

**Patient cohorts and consent.** The UCSF cohort used to develop the RNA-seq classifiers was initially described in our prior study applying metagenomic sequencing to NP swabs from adult patients with mostly mild acute respiratory illnesses tested for COVID-19 early during their disease course (10). Additional samples collected at UCSF since then were sequenced or used for qPCR in the present work. All UCSF samples were collected in accordance with UCSF Institutional Review Board protocol number 17-24056, which granted a waiver of consent. The New York cohort used to validate the RNA-seq classifiers on an external data set was previously published (12).

**RNA-seq data preprocessing.** In the UCSF cohort, we assigned patient samples to one of three viral status groups: (i) samples with a positive clinical RT-PCR test for SARS-CoV-2 were assigned to the "COVID-19" group, (ii) samples with another pathogenic respiratory virus detected by the CZ-ID (formerly, ID-Seq) pipeline (18) in the metagenomic sequencing data were assigned to the "other virus" group, and (iii) remaining samples were assigned to the "no virus" group. The full process for assignment into viral status groups is described in detail in our original study (10), and we applied it as before to the additional swabs reported in the present work.

We wished to retain for classifier development COVID-19 samples with likely active infection (culturable virus), which several studies have related to viral PCR $C_T$ <30 (19 to 21). Because not all $C_T$ values were available, we relied on the relationship between viral reads-per-million (rpM) in the sequencing data and PCR $C_T$ that we previously reported (10): $\log_2(\text{rpM}) = 31.9753 - 0.9167 \cdot C_T$. Metadata for the UCSF samples are provided in Data Set S1.

We pseudoaligned the UCSF samples with kallisto (22) (v. 0.46.1), using the bias correction setting, against an index consisting of all transcripts associated with human protein coding genes (ENSEMBL v. 99), cytosolic and mitochondrial rRNA sequences, and the sequences of ERCC RNA standards. Samples retained in the data set had at least 400,000 estimated counts associated with transcripts of protein coding genes. Gene-level counts were generated from the kallisto transcript abundance estimates using the R package tximport (23) (v. 1.14) with the scaledTPM method. Genes were retained if they had at least 10 counts in at least 20% of samples.

In the New York cohort, samples were also assigned by the authors into the three viral status groups described above based on a combination of RT-PCR and metagenomic sequencing, and we used their assignments as is. Because we did not have access to the underlying sequencing data, we used the gene counts originally generated by the authors using STAR alignment and the R function featureCounts. We excluded samples with less than five million total counts as well as samples that had discordant COVID-19 test results between two assays, but did not filter based on viral load. Genes were retained if they had at least 32 counts in at least 10% of samples. Metadata for the New York samples are provided in Data Set S1.

For each RNA-seq cohort, gene counts were subjected to the variance stabilizing transformation (VST) from the R package DESeq2 (v. 1.26.0), and the transformed values were then standardized (centered and scaled) to yield the final input features.

**RNA-seq SVM classifier development and validation.** SVM learning was implemented in scikit-learn (https://scikit-learn.org) using the sklearn.svm.SVC class function with default parameters and probabilistic output. The UCSF cohort was split into a training set (70%) and a testing set (30%), with stratification to ensure

each set contained a similar proportion of samples with COVID-19. For the greedy feature selection, the performance of a binary SVM classifier to distinguish patients with and without COVID-19 relying on each single feature (gene) was evaluated by running 5-fold cross-validation within the training set and calculating the average AUC across the folds. The three best-performing "first" genes were then selected. To extend these "first" genes to 2-gene combinations, another round of the algorithm was performed, picking the three best-performing "second" genes when combined with each of the "first" genes.

In order to rigorously assess the performance of the SVM 2-gene models, we employed two approaches: (i) running 10,000 rounds of 5-fold cross-validation on the UCSF 70% training set and calculating the average AUC and standard deviation, and (ii) training each model on the UCSF 70% training set and applying it to the 30% testing set to generate an AUC score (Table 1). We then validated the 2-gene models by training each model on the UCSF 70% training set and testing it on the external New York cohort to generate an AUC score (Table 2). Individual sample classification probabilities for the *IFI6*+*GBP5* classifier are tabulated in Data Set S2.

**RT-qPCR of host genes.** RNA was reverse transcribed using the High-Capacity cDNA Reverse Transcription kit (Applied Biosystems), according to the manufacturer's protocol, and analyzed by qPCR in a Bio-Rad CFX384 thermocycler (Bio-Rad) using TaqMan Fast Advanced Master Mix (Applied Biosystems) and TaqMan Gene Expression Assays (Applied Biosystems), according to the manufacturer's protocol. Assay IDs are provided in Table S2. $\Delta C_T$ values were calculated with respect to the reference gene *RPP30* (also known as *RNASEP2*), the standard host control gene used in many viral PCR tests. $\Delta C_T$ values are provided in Data Set S3.

**qPCR SVM classifier proof-of-principle.** The input features for qPCR-based SVM COVID-19 diagnostic classifiers were standardized (centered and scaled) $\Delta C_T$ values. Standardization was performed using the mean and standard deviation of the respective training samples. Performance of the *IFI6*+*GBP5* SVM classifier in distinguishing between the samples with ($n = 72$) and without ($n = 72$) COVID-19 was assessed by 5-fold cross-validation.

We then applied the *IFI6*+*GBP5* classifiers from the 5-fold cross-validation to SARS-CoV-2 variant samples and to samples that had been contaminated with 1:$10^5$ dilution from a high SARS-CoV-2 viral load sample, and calculated the average predicted probability of COVID-19. Because the variant and contamination samples were assayed in separate experiments after the generation of the training data set, they were always processed alongside $n = 6$ to 7 COVID-19 negative controls from the original training data set. The median $\Delta C_T$ difference observed for these control samples between the training data set and the experiment in which they were rerun was applied to all the samples in the respective experiment in order to account for systematic shifts.

**Data availability.** Gene counts for all UCSF samples have been deposited under NCBI GEO accession GSE188678. The New York data set can be obtained according to the Data Availability statement in the original publication (12). Code for RNA-seq and qPCR SVM classifier development and validation is available at https://github.com/czbiohub/Covid-Host-Classifier-Code.

## SUPPLEMENTAL MATERIAL

Supplemental material is available online only.

**DATA SET S1**, XLSX file, 0.04 MB.

**DATA SET S2**, XLSX file, 0.04 MB.

**DATA SET S3**, XLSX file, 0.01 MB.

**FIG S1**, TIF file, 0.5 MB.

**TABLE S1**, DOCX file, 0.02 MB.

**TABLE S2**, DOCX file, 0.01 MB.

**TABLE S3**, DOCX file, 0.01 MB.

## ACKNOWLEDGMENTS

J.A., E.M., J.K., A.O.P. and C.R.L. are listed as inventors on a patent application filed by Chan Zuckerberg Biohub and the University of Califronia San Francisco for the use of host genes for COVID-19 diagnosis.

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
