## [Reviewer comments · mSystems]

A 2-Gene Host Signature for Improved Accuracy of COVID-19 Diagnosis Agnostic to Viral Variants

Jack Albright, Eran Mick, Estella Guerrero, Jack Kamm, Anthea Mitchell, Angela Detweiler, Norma Neff, Alexandra Tsitsiklis, Paula Hayakawa Serpa, Kalani Ratnasiri, Diane Havlir, Amy Kistler, Joseph DeRisi, Angela Pisco, and Charles Langelier

Corresponding Author(s): Charles Langelier, UCSF

Review Timeline:

Submission Date:	July 19, 2022
Editorial Decision:	August 24, 2022
Revision Received:	October 24, 2022
Accepted:	November 1, 2022

Editor: Ileana Cristea

Reviewer(s): The reviewers have opted to remain anonymous.

Transaction Report:

DOI: <https://doi.org/10.1128/msystems.00671-22>

August 24, 2022

Dr. Charles R Langelier
UCSF
Medicine, Division of Infectious Diseases
Chan Zuckerberg Biohub
499 Illinois Street
San Francisco, California 94158

Re: mSystems00671-22 (A 2-Gene Host Signature for Improved Accuracy of COVID-19 Diagnosis Agnostic to Viral Variants)

Dear Dr. Charles R Langelier:

Thank you for submitting your manuscript to mSystems. We have completed our review and I am pleased to inform you that, in principle, we expect to accept it for publication in mSystems. Both reviewers acknowledged the overall quality and value of the study. Several clarifications and additions were requested to further improve the clarity of the manuscript. Acceptance will not be final until you have adequately addressed the reviewer comments.

Preparing Revision Guidelines

Sincerely,

Ileana Cristea

Editor, mSystems

Journals Department
American Society for Microbiology
1752 N St., NW

Reviewer comments:

Reviewer #2 (Comments for the Author):

The manuscript by Albright et al. investigated the utility of a two gene signature in diagnosing COVID-19 from other viral infections and non-viral samples. The gene signature can be validated in external datasets and can be further adapted for RT-qPCR application. The design is clear and the manuscript is well-written.

I have a few comments:

1. In the introduction, the author wrote "Optimal classifiers rely on an interferon-stimulated gene that is strongly induced in COVID-19 compared with non-viral conditions, such as IFI6, and a second immune-response gene that is more strongly induced in other viral infections, such as GBP5". Where is the reference for this observation?
2. IFI6 is an interferon alpha response gene, and GBP5 is involved in interferon gamma signaling. Interferon alpha response comes early in the disease course, while interferon gamma is produced by T cells and the response comes late in the disease stage. Thus, disease course and sample collection time are important factors to consider when evaluating the expression of these two genes. For the samples involved in this study, when are they collected? Are they early or late in the viral infection stage? Is there an imbalance in the sample collection time between COVID-19 and non COVID-19 samples, that may introduce bias in the identification of the signature?
3. Patients with severe outcomes may have impaired interferon response. What is the percentage of this patient group in the datasets, and would this be another source of bias in this study? To put it in another way, if there are more patients with severe outcomes, would the signature still perform well?

Albright et al. present a very interesting study evaluating a support vector machine (SVM) classifier that uses a 2-gene host signature for COVID-19 diagnosis. They identify a need for a diagnostic test that is less vulnerable to potential false-negative PCR results as new SARS-CoV-2 variants emerge, and less vulnerable to false-positive results from cross-contamination in the clinical laboratory. They hypothesize that a SVM using a 2-gene host signature may meet these criteria and performed a proof of concept study with 3 analyses.

1) Retrospective development of a 2-gene classifier: The authors developed the classifier through secondary analysis of data collected for a previously conducted cohort study evaluating upper airway gene expression in patients with SARS-CoV-2 infection (n=90), other viral upper respiratory infections (n=169), and non-viral respiratory illnesses (n=59). The authors identified 9 different 2-gene combinations that best delineated the three groups (COVID, viral infection but COVID, uninfected) with greedy feature selection. They then rigorously tested these SVM classifiers with cross-validation of the training set, application to the test set of this cohort, and application to an external cohort. The authors identified 2 genes that had the best performance (IFI6 and GBP5) as measured by AUC.

2) Prospective evaluation of 2-gene classifier: They measured the expression of the genes that were identified as the best performing 2-gene combination in a new cohort of 73 COVID cases and 73 Upper respiratory illnesses that were not COVID and applied their SVM classifier to this cohort and found that it had an AUC of 0.84.

3) They measured IFI6 and GBP5 in specimens from subjects with Omicron, California N-gene, and Delta variant, as well as samples that were spiked with SARS-CoV-2 but acquired from uninfected subjects. They found that the classifier correctly classified patients with variants as infected and contaminated samples as uninfected.

Overall, I think that this is a very well done and strong proof of concept study. The methods and analyses are sound. My major comment is that the manuscript would be strengthened by better contextualizing how this information would be used. This paper is comparing the classifier **against** the current gold standard – detection of SARS-CoV-2 RNA. Real world use of this assay and classifier would likely be used in conjunction with PCR results and the real test will be what this **adds** to PCR testing, not how it **compares to** PCR testing. This study alone won't be able to tackle this issue since it's proof of concept but I think that some sort of analysis of what viral load is like in misclassified patients is important and would strengthen the manuscript.

Comments

1) While the authors highlight AUC and sensitivity and specificity at Youden's index, what matters clinically is the negative predictive value and the positive predictive value of the test. These are reported in table 1 for the development analysis, but I think some discussion of this in the main text is warranted along with the absolute number of misclassified patients in each cohort. For the prospective evaluation of 73 positive and 73 negative subjects, PPV and NPV aren't reported at all, and they should be.

2) The authors state that the impetus behind the development of this test is to deal specifically with false positive and false negative results from PCR testing. The positive predictive for the IFI6 and GBP5 combination in the external cohort is reported 0.742 (Table1b), which would suggest one out of four patients that are given a diagnosis of COVID-19 by the classifier don't have infection. This would be a

major problem if 2-gene expression SVM classifiers were used alone, but they won't be, they'll be interpreted in the context of a positive PCR result. What's going on with the mis-classified patients? Are there some clinical characteristics that are associated with misclassification? Do they have more severe disease (it appears that the New York cohort had SOFA scores calculated in the original manuscript so I presume that severity of illness would be available to the authors) Is there an association with viral load and probability of misclassification?

Again, this study alone won't be adequate to fully explore this issue but I think some analysis of load in misclassified patients is warranted – this will be a challenge.

3) Even though the Omicron patients look like they would be correctly classified with a probability of infection for most of them at or above 75%, it's striking that Omicron patients have lower probabilities of infection as determined by the classifier compared to delta variant and California N-gene variant. I think that this is interesting biologically as it suggests that host gene expression is different among different variants, and also important clinically as it suggests that even gene expression assays may reach a point where they too are vulnerable to producing false negative results as new variants evolve.

4) Any analysis of new diagnostic testing needs an explicit discussion of what the gold standard for comparison is, and as the manuscript is written right now, this is not easy to find and only referenced in lines 215-216 by directing the reader to the original study manuscripts. I think that summarizing or reporting how patients were classified in this manuscript would make it easier for readers.

5) From the methods of the previous papers cited, I could not determine if these were consecutively collected subjects or a convenience sample. If these were convenience samples, then this could introduce bias into the cohort and should be highlighted as a potential limitation, but if consecutively collected then this is a strength.

Response to Reviewers

Reviewer #1

Albright et al. present a very interesting study evaluating a support vector machine (SVM) classifier that uses a 2-gene host signature for COVID-19 diagnosis. They identify a need for a diagnostic test that is less vulnerable to potential false-negative PCR results as new SARS-CoV-2 variants emerge, and less vulnerable to false-positive results from cross-contamination in the clinical laboratory. They hypothesize that a SVM using a 2-gene host signature may meet these criteria and performed a proof of concept study with 3 analyses.

1) Retrospective development of a 2-gene classifier: The authors developed the classifier through secondary analysis of data collected for a previously conducted cohort study evaluating upper airway gene expression in patients with SARS-CoV-2 infection (n=90), other viral upper respiratory infections (n=169), and non-viral respiratory illnesses (n=59). The authors identified 9 different 2-gene combinations that best delineated the three groups (COVID, viral infection but not COVID, uninfected) with greedy feature selection. They then rigorously tested these SVM classifiers with cross-validation of the training set, application to the test set of this cohort, and application to an external cohort. The authors identified 2 genes that had the best performance (IFI6 and GBP5) as measured by AUC.

2) Prospective evaluation of 2-gene classifier: They measured the expression of the genes that were identified as the best performing 2-gene combination in a new cohort of 73 COVID cases and 73 Upper respiratory illnesses that were not COVID and applied their SVM classifier to this cohort and found that it had an AUC of 0.84.

3) They measured IFI6 and GBP5 in specimens from subjects with Omicron, California N-gene, and Delta variant, as well as samples that were spiked with SARS-CoV-2 but acquired from uninfected subjects. They found that the classifier correctly classified patients with variants as infected and contaminated samples as uninfected.

Overall, I think that this is a very well done and strong proof of concept study. The methods and analyses are sound. My major comment is that the manuscript would be strengthened by better contextualizing how this information would be used. This paper is comparing the classifier against the current gold standard – detection of SARS-CoV-2 RNA. Real world use of this assay and classifier would likely be used in conjunction with PCR results and the real test will be what this adds to PCR testing, not how it compares to PCR testing. This study alone won't be able to tackle this issue since it's proof of concept but I think that some sort of analysis of what viral load is like in misclassified patients is important and would strengthen the manuscript.

We appreciate that the reviewer found our work to represent a strong proof of concept study. We have now added the following Discussion paragraph to provide additional context on how we envision the assay to be used and the type of follow-up studies that would be required, along the lines of the comments above.

Lines 188-195: "Our host-based classifiers were developed and evaluated using the practical gold-standard of clinical viral PCR, which would be more accurate in the general case than any host-based classifier (at least for existing variants). We emphasize, however, that we do not envision the use of host classifiers as a replacement for viral PCR, but rather as a complementary approach to compensate for its potential failure modes. While our proof-of-concept work suggests that addition of host targets is likely to improve overall diagnostic accuracy, a prospective assessment using clinically confirmed false-positive and false-negative viral tests is needed, and a

randomized controlled trial of our assay will be required to firmly establish its clinical utility.”

Comments

1) While the authors highlight AUC and sensitivity and specificity at Youden’s index, what matters clinically is the negative predictive value and the positive predictive value of the test. These are reported in table 1 for the development analysis, but I think some discussion of this in the main text is warranted along with the absolute number of misclassified patients in each cohort. For the prospective evaluation of 73 positive and 73 negative subjects, PPV and NPV aren’t reported at all, and they should be.

We have now added the PPV and NPV metrics of the UCSF and NY cohorts in the main text:

Lines 112-114: “At Youden’s index, the top performing combinations achieved sensitivity in the range of 82-89%, positive predictive value (PPV) as high as 82% and negative predictive value as high as 95%.”

Lines 124-126: “At Youden’s index, the classifiers achieved sensitivity of 83-84%, PPV as high as 74%, NPV as high as 92%, and overall specificity of 88%...”

We note, however, that PPV and NPV depend on the prevalence of the predicted class in each specific cohort. Thus, it is actually the sensitivity and specificity that provide a stable measure of classification performance that can be properly compared across cohorts, and we believe they are the most appropriate to emphasize in a proof-of-concept study such as this.

In addition, we have now provided the PPV and NPV of the qPCR cohort in **Supp. Table 3**. We emphasize that the qPCR assays were intended principally to demonstrate technical feasibility, not as another full-fledged validation cohort.

We now provide in **Supp. Data 2** the classification labels and probabilities for all the samples in the UCSF and NY cohorts using the *IFI6+GBP5* classifier, such that the misclassified samples can be identified. However, we prefer not to overwhelm Table 1 with the absolute numbers of correct or incorrect classifications in each cohort, each 2-gene combination, and each type of analysis. Interested readers could easily calculate these numbers from the sensitivity/specificity metrics in Table 1 and the overall group sizes we report.

2) The authors state that the impetus behind the development of this test is to deal specifically with false positive and false negative results from PCR testing. The positive predictive for the IFI6 and GBP5 combination in the external cohort is reported 0.742 (Table1b), which would suggest one out of four patients that are given a diagnosis of COVID-19 by the classifier don’t have infection. This would be a major problem if 2-gene expression SVM classifiers were used alone, but they won’t be, they’ll be interpreted in the context of a positive PCR result.

We certainly agree with the reviewer’s point. In addition to the Discussion paragraph that we have added (see above), we made sure to indicate throughout the manuscript that the host classifier is meant to be used in conjunction with viral PCR.

Lines 22-24 (Abstract): “Host immune response markers provide an orthogonal indication of infection that can mitigate these concerns when combined with direct viral detection.”

Lines 70-72: "... we address these gaps by identifying 2-gene host signatures that could practically be incorporated into an RT-qPCR (qPCR) assay alongside a control gene and one or more viral targets."

Lines 86-88: "In the present work, we sought to develop a parsimonious 2-gene signature that could be practically incorporated into a PCR assay alongside a control gene and one or more viral targets."

Lines 173-176: "We leveraged multiple cohorts – encompassing over 1,000 patients with COVID-19, other viral ARIs and non-viral conditions – to develop and validate 2-gene host-based COVID-19 diagnostic classifiers that could be practically incorporated into clinical PCR assays in combination with a control gene and one or more viral targets."

What's going on with the mis-classified patients? Are there some clinical characteristics that are associated with misclassification? Do they have more severe disease (it appears that the New York cohort had SOFA scores calculated in the original manuscript so I presume that severity of illness would be available to the authors) Is there an association with viral load and probability of misclassification? Again, this study alone won't be adequate to fully explore this issue but I think some analysis of load in misclassified patients is warranted – this will be a challenge.

We appreciate this important question. Unfortunately, we did not have access to clinical data of patients in the NY cohort and patients in the UCSF cohort were overwhelmingly mild cases, so we were unable to directly examine correlation between disease severity and classification performance. In any case, we envision the host classifier being used principally for initial diagnosis, when most patients are still in the mild stage of disease. Nevertheless, we have added disease severity as a potential modifier of classifier performance in our limitations paragraph:

Lines 196-200: "Our study has some limitations. Our classifier models were trained and tested on cohorts with particular characteristics, including the distribution of COVID-19, other viral, and non-viral cases; the mix of other respiratory viruses represented; and within the COVID-19 group, the distributions of viral load, time since onset of infection and disease severity. Most of these variables likely affect classifier performance and will vary in reality with time and place".

We examined the question of viral load in the NY cohort, which was the largest cohort and had the widest range of viral load. As the reviewer suspected, classification performance was worse in patients with the lowest viral load. We now show this data in **Supp. Fig. 1**:

Supplementary Figure 1. Scatter plot showing the probability of COVID-19 based on the *IFI6+GBP5* RNA-seq classifier as a function of SARS-CoV-2 viral load (C_t) for COVID-19 patients in the NY cohort ($n=166$). Low = $C_t < 18$, Medium = C_t 18-24, High = $C_t > 24$.

And we refer to it in the text, as follows:

Line 126-127: “We observed that COVID-19 samples with very low viral loads were more likely to be misclassified as negative for COVID-19 (**Supp. Figure 1**).”

3) Even though the Omicron patients look like they would be correctly classified with a probability of infection for most of them at or above 75%, it’s striking that Omicron patients have lower probabilities of infection as determined by the classifier compared to delta variant and California N-gene variant. I think that this is interesting biologically as it suggests that host gene expression is different among different variants, and also important clinically as it suggests that even gene expression assays may reach a point where they too are vulnerable to producing false negative results as new variants evolve.

We now discuss the possibility that host gene expression could vary based on SARS-CoV-2 variant, potentially reflecting mechanisms of immune evasion or unique features of the host immune response.

Lines 204-206: “Finally, it is possible that host gene expression would differ in response to infection with future SARS-CoV-2 variants, which could impact host-based diagnosis.”

4) Any analysis of new diagnostic testing needs an explicit discussion of what the gold standard for comparison is, and as the manuscript is written right now, this is not easy to find and only referenced in lines 215-216 by directing the reader to the original study manuscripts. I

think that summarizing or reporting how patients were classified in this manuscript would make it easier for readers.

We appreciate this suggestion and have added the key information on how patients were assigned to viral status groups in the main text:

Lines 90-94: "The full UCSF cohort used in the present work included n=318 patients, of whom 90 had PCR-confirmed COVID-19 (with viral load equivalent to PCR Ct < 30), 59 had other viral infections detected by metagenomic sequencing (mostly rhinovirus and influenza), and 169 had no virus detected and were presumed to suffer from non-viral ARIs."

More details are provided in the Methods section:

Lines 228-234: "In the UCSF cohort, we assigned patient samples to one of three viral status groups: 1) samples with a positive clinical RT-PCR test for SARS-CoV-2 were assigned to the "COVID-19" group, 2) samples with another pathogenic respiratory virus detected by the CZ-ID (formerly, ID-Seq) pipeline in the metagenomic sequencing data were assigned to the "other virus" group, and 3) remaining samples were assigned to the "no virus" group. The full process for assignment into viral status groups is described in detail in our original study, and we applied it as before to the additional swabs reported in the present work."

5) From the methods of the previous papers cited, I could not determine if these were consecutively collected subjects or a convenience sample. If these were convenience samples, then this could introduce bias into the cohort and should be highlighted as a potential limitation, but if consecutively collected then this is a strength.

We would like to clarify that the analyzed nasopharyngeal swabs were a combination of a convenience sample derived from a clinical SARS-CoV-2 testing laboratory, as well as publicly available data. We now note this as a limitation as follows:

Lines 200-203: "Moreover, the analyzed nasopharyngeal swabs represented a convenience sample derived from a clinical SARS-CoV-2 testing laboratory, as well as from publicly available data, which could introduce bias."

Reviewer #2

The manuscript by Albright et al. investigated the utility of a two gene signature in diagnosing COVID-19 from other viral infections and non-viral samples. The gene signature can be validated in external datasets and can be further adapted for RT-qPCR application. The design is clear and the manuscript is well-written.

I have a few comments:

1. In the introduction, the author wrote "Optimal classifiers rely on an interferon-stimulated gene that is strongly induced in COVID-19 compared with non-viral conditions, such as IFI6, and a second immune-response gene that is more strongly induced in other viral infections, such as GBP5". Where is the reference for this observation?

We would like to clarify that this statement in the Abstract refers to our own findings in the present study. We have reworded this sentence, as follows:

Lines 27-29 (Abstract): “We find that optimal classifiers include an interferon-stimulated gene that is strongly induced in COVID-19 compared with non-viral conditions, such as IFI6, and a second immune-response gene that is more strongly induced in other viral infections, such as GBP5.”

2. IFI6 is an interferon alpha response gene, and GBP5 is involved in interferon gamma signaling. Interferon alpha response comes early in the disease course, while interferon gamma is produced by T cells and the response comes late in the disease stage. Thus, disease course and sample collection time are important factors to consider when evaluating the expression of these two genes. For the samples involved in this study, when are they collected? Are they early or late in the viral infection stage? Is there an imbalance in the sample collection time between COVID-19 and non COVID-19 samples, that may introduce bias in the identification of the signature?

The majority of UCSF samples were collected early during disease course, at the time of the first COVID-19 PCR test. We have clarified this in the methods, as follows:

Line 219-221: “The UCSF cohort used to develop the RNA-seq classifiers was initially described in our prior study applying metagenomic sequencing to NP swabs from adult patients with mostly mild acute respiratory illnesses tested for COVID-19 early during their disease course.”

Nevertheless, we recognize that disease severity and time since onset of infection are potentially important modifiers of host gene expression and have noted this in our limitations paragraph:

Lines 196-200: “Our study has some limitations. Our classifier models were trained and tested on cohorts with particular characteristics, including the distribution of COVID-19, other viral, and non-viral cases; the mix of other respiratory viruses represented; and within the COVID-19 group, the distributions of viral load, time since onset of infection and disease severity. Most of these variables likely affect classifier performance and will vary in reality with time and place”.

3. Patients with severe outcomes may have impaired interferon response. What is the percentage of this patient group in the datasets, and would this be another source of bias in this study? To put it in another way, if there are more patients with severe outcomes, would the signature still perform well?

We would like to note, as above, that the majority of patients in the UCSF cohort had mild disease and were enrolled during an early stage of infection. While critically ill COVID-19 patients may have impaired interferon responses late in their disease course when viral loads are low, differences in interferon signaling are subtle during early stages of infection (e.g., Ng et al. Science Advances, 2021). Given that our test is designed to diagnose patients during early stages of infection, such minor differences would not be expected to dramatically alter test performance. However, definitively answering this question would necessitate prospective evaluation in patients with severe disease who are early during their course of infection. As noted above, we have added disease severity as another potential modifier of classifier performance in our limitations paragraph.

November 1, 2022

Dr. Charles R Langelier
UCSF
Medicine, Division of Infectious Diseases
Chan Zuckerberg Biohub
499 Illinois Street
San Francisco, California 94158

Re: mSystems00671-22R1 (A 2-Gene Host Signature for Improved Accuracy of COVID-19 Diagnosis Agnostic to Viral Variants)

Dear Dr. Charles R Langelier:

Thank you for submitting your revised manuscript to mSystems, and for fully addressing the reviewers' comments.

I am delighted to let you know that your manuscript has now been accepted for publication. Congratulations! I am forwarding it to the ASM Journals Department for publication. For your reference, ASM Journals' address is given below. Before it can be scheduled for publication, your manuscript will be checked by the mSystems production staff to make sure that all elements meet the technical requirements for publication. They will contact you if anything needs to be revised before copy editing and production can begin. Otherwise, you will be notified when your proofs are ready to be viewed.

Publication Fees:

If you would like to submit a potential Featured Image, please email a file and a short legend to msystems@asmusa.org. Please note that we can only consider images that (i) the authors created or own and (ii) have not been previously published. By submitting, you agree that the image can be used under the same terms as the published article. File requirements: square dimensions (4" x 4"), 300 dpi resolution, RGB colorspace, TIF file format.

We recognize that the video files can become quite large, and so to avoid quality loss ASM suggests sending the video file via <https://www.wetransfer.com/>. When you have a final version of the video and the still ready to share, please send it to mSystems staff at msystems@asmusa.org.

Sincerely,

Ileana Cristea
Editor, mSystems

Journals Department
Supplemental Material: Accept
Supplementary Table 3: Accept
Supplemental Material: Accept
Supplementary Table 2: Accept
Supplementary Table 1: Accept
Supplementary Figure 1: Accept
Supplemental Material: Accept